# Estimating psychophysiological loads by repeated temperature steps on humans using a state–space model

Miho Iwasaki[1,2], Yusuke Morito[1], Kyosuke Watanabe[1,3], Kiyoshi Kuroi[1,4], Shota Hori[1,4], Yoko Sakata[1,4], Kei Mizuno[1,3], Kazunobu Okazaki[1,2], Yasuyoshi Watanabe[1,3]*

**1** RIKEN Center for Biosystems Dynamics Research, Chuo-ku, Kobe City, Hyogo, Japan, **2** Department of Environmental Physiology for Exercise, Osaka Metropolitan University Graduate School of Medicine, Osaka City, Japan, **3** Kobe University Graduate School of Science, Technology and Innovation, Chuo-ku, Kobe City, Hyogo, Japan, **4** DAIKIN Industries. Ltd. Co. Technology and Innovation Center, Settsu City, Osaka, Japan

* yywata@riken.jp

## Abstract

Humans are exposed to daily temperature differences indoors and outdoors worldwide; however, the associated risks to health and fatigue remain unclear. This study aimed to clarify the psychophysiological loads by repeated short-term temperature differences on Japanese individuals. Herein, 28 healthy individuals were repeatedly moved between two temperature environments, and their psychological/physiological responses to temperature differences in the environment were recorded [$T_{26-26}$ (control), $T_{26-31}$ (5 °C step), $T_{26-36}$ (10 °C step), and $T_{21-36}$ (15 °C step)]. We precisely estimated the accumulated effects (load) of repeated temperature steps using a Bayesian state–space model, and distinguished them from the direct effects of environmental changes. The Load to the autonomic nervous system was continuously enhanced (decreased high-frequency of RRI and increased low-frequency/high-frequency of RRI) in the trials with temperature steps, while it was less under the $T_{21-36}$ (15 °C step) than under the $T_{26-36}$ (10 °C step) condition. These findings could help formulate fatigue management approaches and recommend best practices to minimise adverse health effects related to sudden and uncontrollable environmental temperature steps/changes in everyday scenarios on the public.

## Introduction

Japan has four seasons wherein large seasonal temperature changes occur, ranging from a maximum temperature of 36 °C (96.8 °F) in summer and a minimum temperature of 1 °C (33.8 °F) in winter at Kobe City. In recent years, global warming and the heat island effect [1] have caused daytime outdoor temperatures to rise during the summer months, while the development of air conditioning technology has facilitated the control of indoor temperatures to a comfortable level. A large-scale study by

**Data availability statement:** All data necessary for reproducing the figures presented in the manuscript and evaluating the performance of Bayesian estimation/prediction models are included in Supporting Information files.

**Funding:** Y.W. received funding for this study from DAIKIN Industries, Ltd. Co. The funder had no role in study design, data collection and analysis, decision to publish, or preparation of the manuscript. The funder provided support in the form of salaries for authors K.K., S.H., and Y.S., but did not have any additional role in the study design, data collection and analysis, decision to publish, or preparation of the manuscript.

**Competing interests:** I have read the journal's policy and the authors of this manuscript have the following competing interests: This study was conducted under a collaborative research agreement with DAIKIN Industries, Ltd. Y.W. received funding for this study from DAIKIN Industries, Ltd. Co. The funder had no role in study design, data analysis, decision to publish, or preparation of the manuscript. The funder provided support in the form of salaries for authors K.K., S.H., and Y.S., but did not have any additional role in the study design, data analysis, decision to publish, or preparation of the manuscript. Accordingly, we revised the author contribution section in the revised version. The other authors have declared that no competing interests exist. This does not alter our adherence to PLOS ONE policies on sharing data and materials.

Cao et al. [2] found that individuals who use indoor air conditioning in daily life are at a higher risk of subjective discomfort like 'fatigue, weakness or drowsiness', 'eye dryness, itching or tearing', and so on. Previous studies of short-term exposure to temperature steps (temperature differences, like temperature gaps between indoor and outdoor) also indicates psychological and physiological effects in humans [3–10]. Xiong et al. shows that large temperature changes cause subjective symptoms such as perception of sweating, eye strain, and dizziness [3]. Xiong et al. also showed that serum level of interleukin-6, oral temperature, skin temperature, heart rate (HR), and heart rate variability (HRV) [time-domain parameters, e.g., root mean square of successive differences; and frequency-domain parameters, e.g., low-frequency component/high-frequency component (LF/HF)] were sensitive to temperature step changes, and further suggested that both the intensity and direction of the temperature step have a significant effect on human physiological parameters [4]. However, it is unclear what effects repeated short-term exposure to temperature differences will have on the mind and body. Currently, most of the temperature step-load studies have involved only one step and have mainly investigated the effects of single-shot temperature differences [3–10]. Individuals become more tolerant to heat when alternating between sunlight and shade outdoors during the summer; however, the physiological aspects were not investigated [11]. The effects of repeated exposure to high- and low-temperature environments on physical and mental fatigue remain unknown. Clarifying the psychophysiological effects of such exposure is important to maintain and promote human health, though minimise adverse health effects related to sudden and uncontrollable environmental temperature steps/changes in daily life.

Among the research results of the temperature step, HRV indicates a person's state of fatigue through the autonomic nervous system. Recent studies suggest that autonomic nerve dysfunction and fatigue are closely related, with high fatigue leading to increased sympathetic nerve activity and decreased parasympathetic nerve activity [12,13].

The purpose of this study was to observe the psychological and physiological effects of multiple consecutive temperature steps and to quantify the effects of repeated temperature steps on humans. We believe that the measured psychological and physiological parameters include a mixture of immediate effects caused by environmental factors and the effects of this accumulated load. We used a Bayesian state–space model, which is a widely accepted method for a time series data analysis to separate and evaluate load accumulation individually from psychophysiological indicators that involve complex factors. Furthermore, we attempted to predict future fluctuations in autonomic nerve activity using measured values in environments with repeated temperature steps using a Bayesian state–space model. The use of this technique to estimate the Load of future temperature steps could be useful to evaluate anti-fatigue solutions for the Loads of heat and temperature steps.

## Methods

### Participants

Twenty-eight healthy individuals (14 females and 14 males) with a mean age (SD) of 44.2 (9.3) years participated in the study. Table 1 provides a summary of their

**Table 1. Anthropometric information of participants in this study.**

| Sex | Number | Age (years) | Height (cm) | Weight (kg) |
|---|---|---|---|---|
| Female | 14 | 46.4±9.7 | 161.8±9.0 | 55.8±2.5 |
| Male | 14 | 42.0±8.6 | 172.2±13.0 | 71.2±4. |

Data are presented as mean±SD.

anthropometric information. The participants were free from cardiovascular diseases, respiratory diseases, autonomic dystonia, psychological disorders, diabetes, and skin diseases. They wore short-sleeved T-shirts, sweatpants, socks, and slippers supplied by the experimenter during the experiment. All the participants were asked to avoid alcohol the day before each experiment and caffeine in the morning of each experiment and finished their meal 1 h prior to each experiment.

The study protocol was approved by the Ethics Committee of RIKEN (Kobe2 2018−01) and conducted in accordance with the Declaration of Helsinki. All the participants provided written informed consent prior to enrolment. Participants were recruited from August 9, 2019, to August 30, 2019, for this study.

## Experimental conditions

The experiment was conducted in a climate chamber with two adjacent rooms (A and B) connected by internal doors. We set the temperature steps to $T_{26-26}$ (unit °C; 78.8–78.8 ℉, control condition), $T_{26-31}$ (5 °C difference, 78.8–87.8 ℉), $T_{26-36}$ (10 °C difference, 78.8–96.8 ℉), $T_{21-36}$ (15 °C difference, 69.8–96.8 ℉, although the base is 21 °C instead of 26 °C, cool condition). Room A was set at temperatures of 21 °C or 26 °C, and room B was set at temperatures of 26 °C, 31 °C, or 36 °C. The relative humidity was set at 50% in both rooms. We set the baseline temperature to 26°C because a previous study suggested that an indoor temperature at approximately 26–27°C was comfortable in summer [14]. The Japanese Ministry of Health, Labor and Welfare uses WBGT (Wet Bulb Globe Temperature) values to evaluate the risk of heatstroke [15]. A room temperature of 36 °C with a humidity of 50% has a WBGT value of 31, which is considered to be the upper limit for heatstroke risk. For this reason, 36 °C was set as the maximum room temperature for this study. We used a temperature of 31 °C (the median between 36 °C and 26 °C) to compare the effects of multiple temperature differences.

## Measurements

We conducted psychological measurements using a visual analogue scale (VAS). The participants were asked to indicate their perception of physical fatigue on a scale of 0 (no fatigue) to 100 (complete exhaustion). The VAS has been widely used in fatigue research and is supported by evidence of high reliability and validity in capturing subjective fatigue levels [16]. Its continuous nature allows for fine-grained assessment of subtle changes in fatigue that might be missed by categorical or Likert-type scales, and is particularly useful for tracking temporal variations in fatigue states over time. Physiological measurements included HR, HRV, and skin temperature. All the physiological measurement instruments were attached to the participants during the experiment, and they were asked not to touch them. Their electrocardiogram (ECG) readings were recorded using a portable recorder (Bonaly Light, GMS Co., Ltd., Tokyo, Japan), which was attached to the participants' chest. The HR and HRV were computed based on the ECG readings. The skin temperature of four body parts (namely, the chest, thighs, upper arm, and leg) was logged every second using a portable recorder (LT-8, Gram Co., Ltd., Saitama, Japan). The mean skin temperature was calculated according to the following equation [17]:

$$T_{skin} = 0.3 * T_{Chest} + 0.3 * T_{Upperarm} + 0.2 * T_{thigh} + 0.2 * T_{leg}$$

 

## Experimental procedure

The experiments were conducted from 20 August to 30 September 2019. The detailed protocol is shown in Fig 1. Participants visited the site for four days and participated in one experimental condition per day. They participated in the next condition at least 2 days after participating in one condition. The order of conditions was counterbalanced. Before each test, the participants changed their clothes to short sleeves and long trousers and the experimenter set them up with physiological measurement instruments in the pre-test area. Subsequently, the participants walked into room A and sat on a chair for 6 min (Preparation section). Then, the experiment was initiated. First, the individuals stayed in room A for 14 min (Room $A_{pre}$). They were then moved to room B and stayed there for 21 min, returned to room A and stayed there for 21 min, and moved to room B and stayed there for 21 min. The participants repeated this sequence four times. Finally, the participants stayed in room A for 14 min (Room $A_{post}$). The total time for one test was 183 minutes. To exclude the effect of exercise (such as walking) during the experiment, the participants remained seated at all times, and the experimenters transferred the seated participants on casters. At the first and last session in room A, the participants were asked to close their eyes and rest in their chair to accurately record their ECG in a rest state.

The experiment was designed for the participants to repeatedly move back and forth between rooms A (without heat load) and B (with heat load, except control condition). Participants acclimatised, rested, answered the questionnaire, and then moved on. The participants then went back and forth between rooms B and A four times, repeating the questionnaire and resting. Participants returned to room A at the end, rested, completed the questionnaire, and the experiment was completed. Overall, the experiment lasted 183 min.

The first questionnaire section took 4 min wherein the participants could practice answering the questionnaires, which including questions related to the VAS and their physical and mental conditions. Physical and mental conditions were recorded for the experimenters to monitor the participants' safety. The basic questionnaire sections took 1 min to complete. In the 3 min questionnaire section, the participants answered questions on their physical and mental conditions in addition to the VAS.

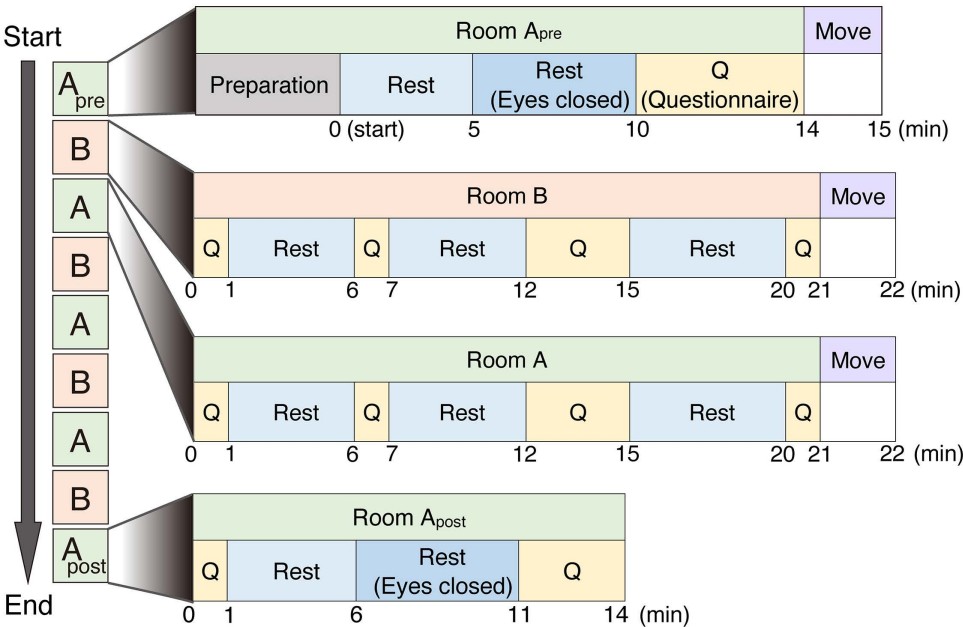

**Fig 1. Schematic representation of the detailed study protocol.**

## Statistical analysis

The HR and HRV parameters were calculated from ECG data. The high frequency (HF), and low frequency (LF)/HF were calculated by Maximum Entropy Method [18] from the ECG data. The HF is the 0.15–0.4 Hz band and LF is the 0.04–0.15 Hz bands of the R-R Interval (RRI). The HF was log calculated, and LF/HF was calculated using the logs of LF and HF. ECG was recorded continuously throughout the experiment, but HR and HRV were evaluated only during the Rest section. Each Rest section was five minutes long, but the first 30 seconds were excluded from the calculation because they may have been affected by the previous section. HR and HRV were calculated and averaged for the remaining four minutes and 30 seconds, and this data was used as one time point. There were 23 time points of HR and HRV data in the entire time series of one experiment. Skin temperature was also recorded continuously throughout the experiment. Skin temperature data was divided into three equal time ranges and averaged for the first room $A_{pre}$ and the last room $A_{post}$, and divided into four equal time ranges and averaged for rooms A and B. In total, there were 34 time points of data of skin temperature. The VAS was recorded in the questionnaire section, and there were 31 time points of data.

## State–space model

We applied a state–space model with Bayesian statistical estimates. We hypothesized that the psychophysiological index contains three components and created a model formula using Bayesian statistics to individually evaluate the accumulation of load on the mind and body. We defined the three components as 'Load', 'Base', 'Environment (Env)'. "Load" is defined as the accumulation of burden on the mind and body owing to repeated exposure to temperature steps. 'Base' indicates the effect other than the temperature steps. Humans may feel some kind of fatigue owing to the passage of time even in a resting state, such as sitting or waiting. Since these factors are common to all conditions, a variable called 'Base' was defined. 'Env' reflects direct effects of temperature differences that occur immediately owing to changes in the environment. The observation equation and state equation in Bayesian statistical estimates are defined as follows:

### Observation equations

$$T_{26-26} = Base + \varepsilon_a$$

$$T_{26-31} = Base + Load_b + Env_b + \varepsilon_b$$

$$T_{26-36} = Base + Load_c + Env_c + \varepsilon_c$$

$$T_{21-36} = Base + Load_d + Env_d + \varepsilon_d$$

$$\varepsilon_n \sim normal\,(0, \sigma_{n^2})$$

### State equations

$$Base_t = 2 * Base_{t-1} - Base_{t-2} + \varepsilon_{Base,t-2}$$

$$\varepsilon_{Base,t} \sim normal\,(0,\ \sigma_{Base})$$

$$Load_{t,n} = 2 * Load_{t-1,n} - Load_{t-2,n} + \varepsilon_{Load,t-2}$$

$$\varepsilon_{Load,t} \sim Cauchy\,(0,\ \sigma_{Load})$$

$$Env_{t,n} = -h \sum_{t=1}^{N-1} Env_{t,n} + \varepsilon Env_{t,n}$$

$$\varepsilon Env_{t,n} \sim normal\,(0,\ \sigma_{Env})$$

Where '$t$' is each time point, and '$n$' refers to each condition ($a = T_{26\text{-}26}$, $b = T_{26\text{-}31}$, $c = T_{26\text{-}36}$, and $d = T_{21\text{-}36}$). In the observation equation, $T_{26\text{-}26}$ to $T_{21\text{-}36}$ are the observed data. Functions of hidden state variables 'Base', 'Load', and 'Environment' define the observed data. $\varepsilon_n$ is the observation noise. $\varepsilon_n$ follows a normal distribution with mean 0 and standard deviation σ. In the state equations, hidden state variables are defined. 'Base' and 'Load' are modelled with second-order difference equations. 'Environment' involves the direct effect of temperature steps and is a seasonal variable owing to regularly repeated step changes in temperature. The state noise $\varepsilon_{Load}$ follows Cauchy distribution, assuming that the difference in load effect is large between the subjects. The other state noise follows a common normal distribution.

In this study, we also performed prediction. In the prediction model, the measured values up to the third set were used to estimate variations in the fourth set. The state equation for the fourth set of prediction intervals is as follows:

State equations for prediction model

$$Base_t = 2 * Base_{t-1} - Base_{t-2} + \varepsilon_{Base,t-2}$$

$$\varepsilon_{Base,t} \sim normal\,(0,\ \sigma_{Base})$$

$$Load_{t,n} = 2 * Load_{t-1,n} - Load_{t-2,n} + \varepsilon_{Load,t-2}$$

$$\varepsilon_{Load,t} \sim Normal\,(0,\ \sigma_{Load})$$

$$Env_{t,n} = - \sum_{t=1}^{N-1} Env_{t,n} + \varepsilon Env_{t,n}$$

$$\varepsilon_{Env,t} \sim Normal\,(0,\ \sigma_{Env})$$

where '$t$' stands for each time point, and '$n$' refers to each condition ($a = T_{26\text{-}26}$, $b = T_{26\text{-}31}$, $c = T_{26\text{-}36}$, and $d = T_{21\text{-}36}$).

We performed a parameter estimation using the framework of Bayesian statistical inference with Markov Chain Monte Carlo (MCMC) methods implemented in CmdStan v. 2.26.1 (https://mc-stan.org) with Cmdstanr v. 2.29.2 (https://mc-stan.org/cmdstanr/).

Regarding the HRV index of one participant, there is a missing value at the first time point owing to a communication error of the measuring equipment. However, a model estimation can be performed even if there is a missing value with Bayesian statistical modelling. No other data were missing.

In Bayesian statistics, a 95% credible interval is "the interval in the posterior probability distribution of the population mean that contains the true value with 95% probability". In the study, the absence of overlap in the 95% credible interval was considered as the statistically significant difference, and we discussed the difference by using Bayesian credible interval. In this study, we have focused on and discussed the 'Load' model, which is the accumulation of mental and physical loads owing to temperature steps.

## Results

### Bayesian statistical inference with Markov Chain Monte Carlo (MCMC)

Four independent MCMC chains were run with 2,000 steps each for the warm-up and 2,000 steps for the sampling iterations. We confirmed that all the estimated parameters had < 1.05 R∘ convergence diagnostic and more than 1,000 effective sampling size (ESS) values, indicating that the MCMC runs were successfully convergent. R^ is a convergence diagnostic statistic and values less than 1.1 indicate adequate convergence [19]. The ESS is a measure for sampling efficiency and values greater than 400 indicate that the estimates are reliable [20]. Taken together, the estimates of each model in this study are reliable. The details of the parameters are provided in the Supplementary Information (S1–S10 Tables).

### Subjective index and skin temperature

Fig 2 shows the subjective index and skin temperature results (the corresponding data and model parameters are provided in S1, S2 and S11 Tables). In the subjective index, the higher the value, the greater the feeling of physical fatigue. Feeling of physical fatigue tends to increase with repetition under all conditions, including the $T_{26-26}$ control condition (Fig 2a). Regarding the Load, the 50% credible intervals of conditions $T_{21-36}$, $T_{26-36}$ and $T_{26-31}$ did not overlap, and condition $T_{21-36}$ tended to have the highest subjective physical fatigue, while condition $T_{26-31}$ tended to have the lowest estimated value.

The skin temperature of the model and data overlap (Fig 2b, left). The skin temperature fluctuated depending on the room temperature but gradually fell under the $T_{26-26}$ condition where the room temperature did not change (Fig 2b). $T_{21-36}$ was significantly lower than the $T_{26-31}$ and $T_{26-36}$ conditions considering the Load at the first time point. Meanwhile, the $T_{26-36}$ condition was significantly higher than the other conditions from the moment participants first enter Room B (the fourth time point) to the last point.

### Heart rate variability

The measured values and estimated models of HF gradually increased under the $T_{26-26}$ condition, while the Load gradually decreased under the condition with a temperature step ($T_{21-36}$, $T_{26-36}$, and $T_{26-31}$) (Fig 3a). The $T_{21-36}$ condition tended to be higher than the other conditions without overlap of the 50% credible interval at the first time point of the Load of HF. Finally, the $T_{26-36}$ condition tended to be lower with no overlapping 50% credible intervals with the other conditions.

In the Load of LF/HF, the 50% credible intervals for $T_{21-36}$ condition did not overlap with the other conditions and tended to be lower than the other conditions at the first time point, although those for the $T_{21-36}$ and $T_{26-31}$ conditions overlap from the seventh point (at the last Rest section when participants entered Room A for the second time). From the 12th point (at the middle Rest section when participants entered Room A for the third time), the $T_{26-36}$ condition showed a higher trend with no 50% credible interval overlap with the other conditions.

The Load of HR showed no difference at the first time point owing to the overlap of the 95% credible intervals. Finally, the lack of overlap of the 50% credible intervals showed a trend towards higher $T_{26-36}$, $T_{21-36}$, and $T_{26-31}$, in that order. The data and model parameters of HF, LF/HF, and HR are provided in S11,S3,S4 and S5 Tables.

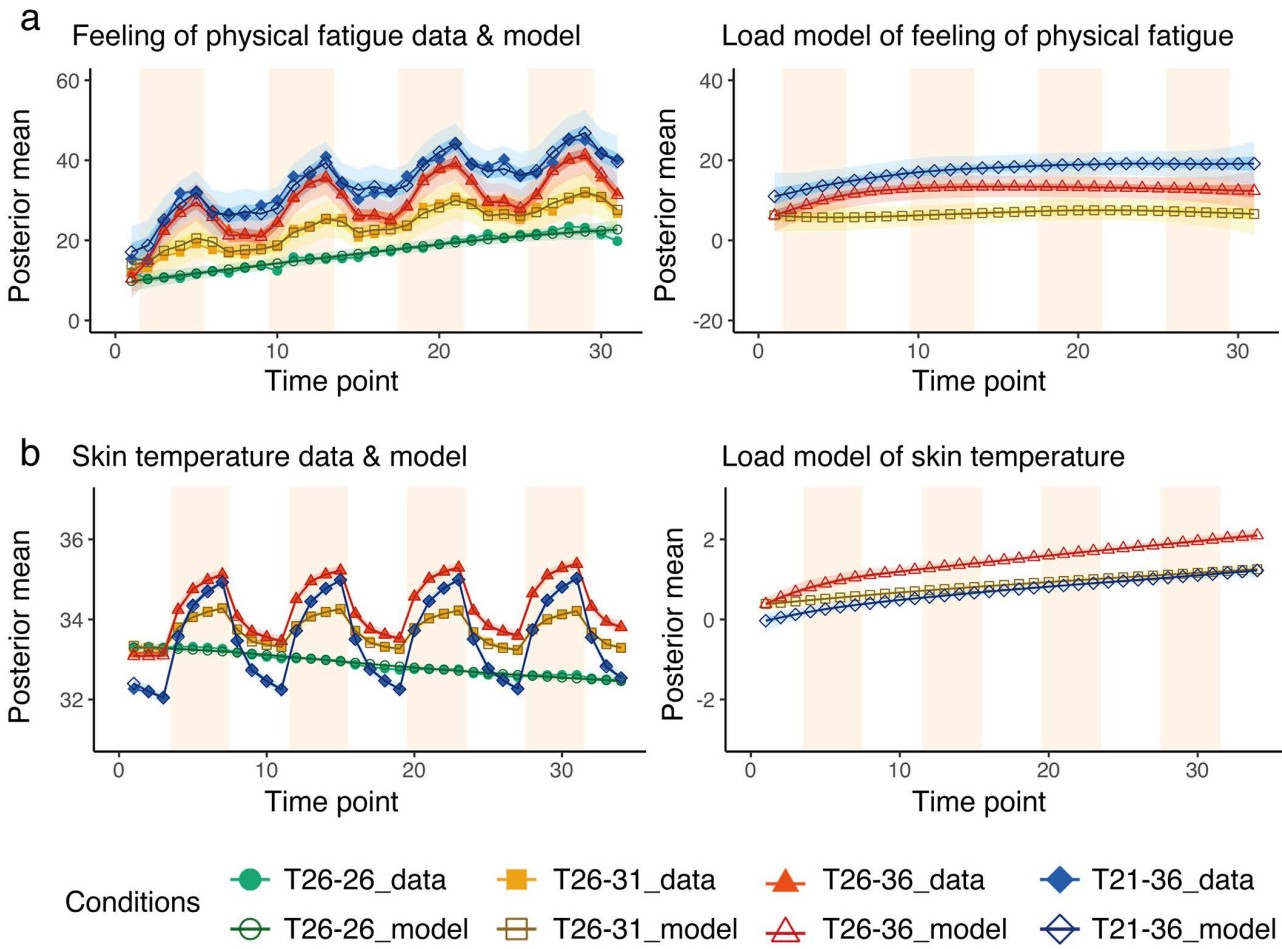

**Fig 2. Subjective index and skin temperature.** Results of (a) subjective physical fatigue and (b) skin temperature modelling. The left charts show the superimposed model estimates and mean values of the actual measurements for 28 participants. The right charts show the Load term, which is the accumulated load. In the left charts, empty markers represent the posterior average of the modelling, and filled markers represent the average of the actual measured data for 28 participants. The light band represents the 95% credible interval, and the dark band represents the 50% credible interval for each condition. The light orange background indicates room B. The x-axis shows the time points, and the y-axis shows the measured and model-estimated values (posterior mean) of each index.

## Prediction results

For prediction model, the 'Base', 'Load', and 'Environment' models for all the conditions were estimated with R^ less than 1.05 and ESS greater than 1,000. In the prediction model, the measured values up to the third set were used to estimate variations in the fourth set (Fig 4 and 5; the corresponding data and prediction model parameters are provided in S11,S6,S7,S8,S9 and S10 Tables.

Fig 4 and 5 show the measured values and prediction models. The models to the left of the black vertical line are estimated based on actual measurement values at the corresponding time points. The predicted values from the fourth set onward, which are to the right of the black vertical line, are estimated using the model based on actual measurements up to the third set. The coloured band represents the 50% credible interval.

The credible interval for predicting physical fatigue expanded, with a tendency toward overestimating fatigue (Fig 4a). Measured values and prediction models overlapped to a certain degree for skin temperature (Fig 4b). Measured values

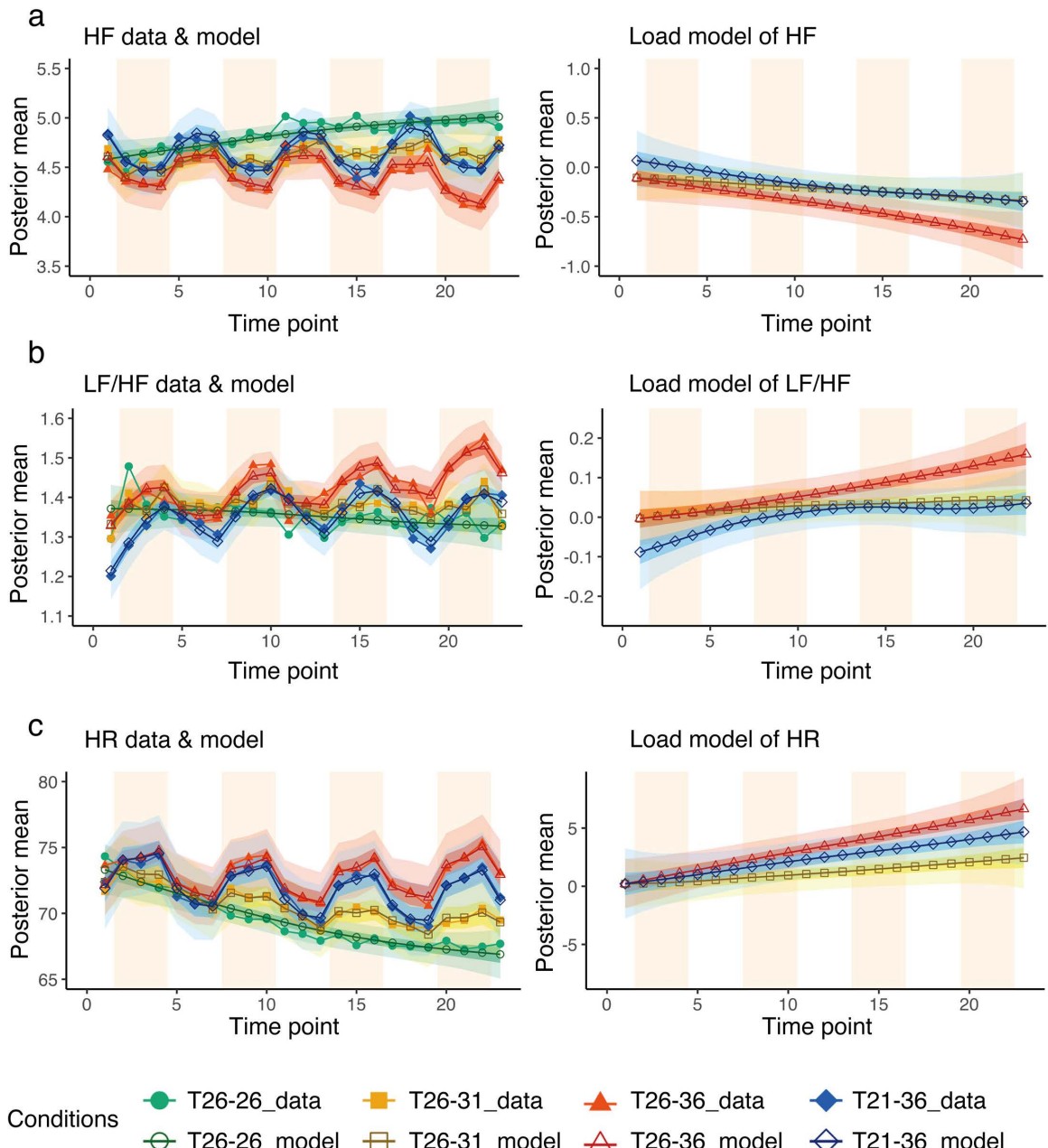

**Fig 3. Heart rate variability.** Results of (a) HF (high-frequency component of heart rate variability), (b) LF/HF (the ratio between low- frequency component and high-frequency component of heart rate variability), and (c) HR (heart rate) modelling. The left charts show the superimposed model estimates and mean values of the actual measurements for 28 participants. The right charts show the Load term, which is the accumulated load. In the left charts, empty markers represent the posterior average of the modelling, and filled markers represent the average of the actual measured data for 28 participants. The light band represents the 95% credible interval, and the dark band represents the 50% credible interval for each condition. The light orange background indicates room B. The x-axis shows the time points, and the y-axis shows the measured and model-estimated values (posterior mean) of each index.

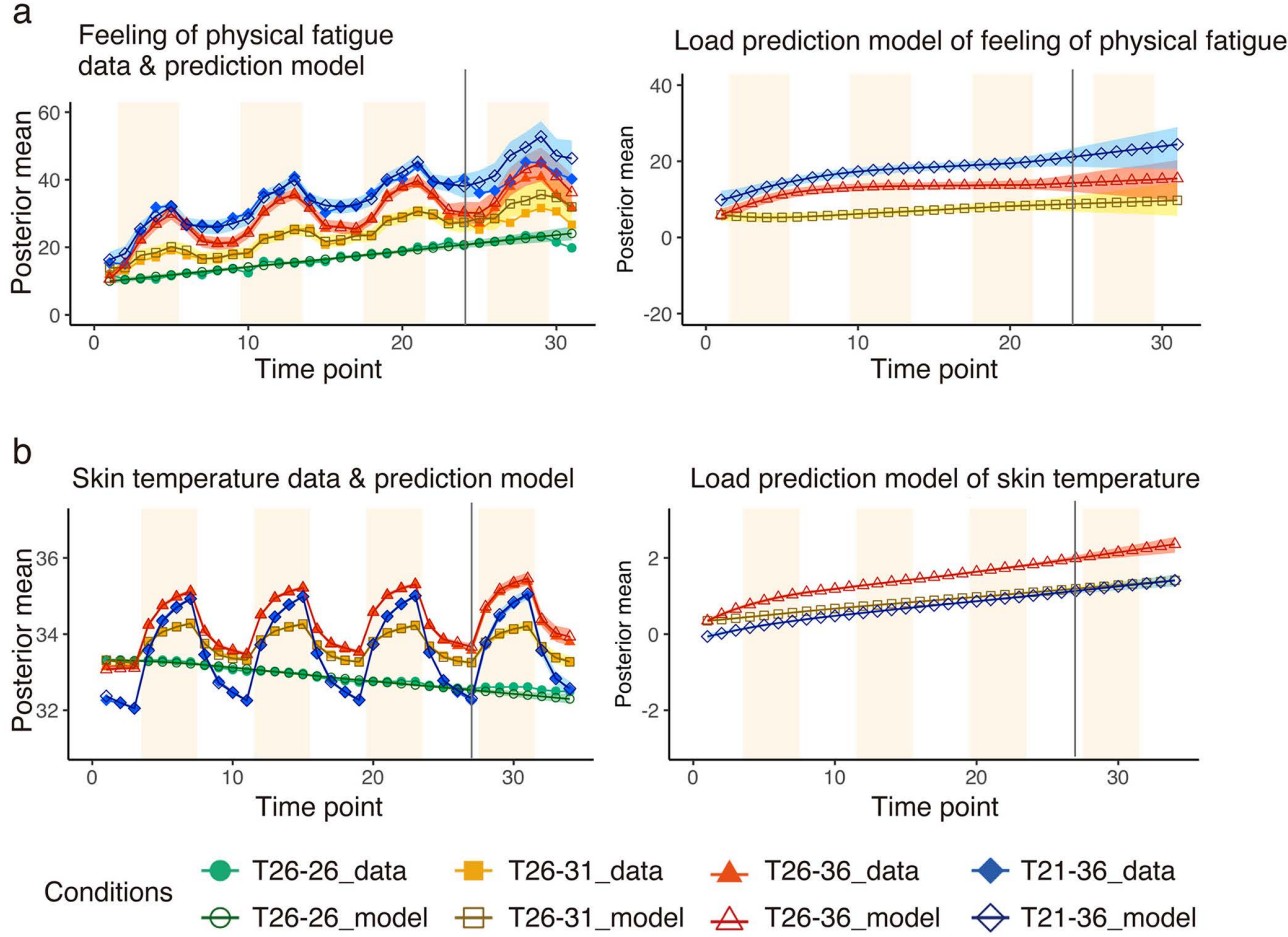

**Fig 4. Prediction of subjective index and skin temperature.** Results of the (a) subjective physical fatigue and (b) skin temperature predictions. Model estimates to the left of the vertical black line are based on actual measurements at the corresponding time points, and the models after the fourth set on the right side of the black vertical line are estimated using the measured values up to the third set. The colour band represents the 50% credible interval. The light orange background indicates room B. The x-axis shows the time points, and the y-axis shows the measured value, estimated model, and predicted model of each index. Filled markers represent the measured values, and empty markers represent the estimated models.

and prediction models overlapped to some degree in the $T_{26-36}$ condition for HRV, whereas there were some differences in the other conditions (Fig 5).

## Discussion

We used Bayesian statistical modelling to extract and visualise the accumulation of latent psychological and physiological loads contained in the data. The Load of HF (which is an index of parasympathetic nerve activity) gradually decreased, while the index of sympathetic nerve activity (LF/HF) increased when repeatedly receiving temperature steps. The rise and fall widths of the Load in HF and LF/HF were reduced under the $T_{21-36}$ condition compared to those observed under the $T_{26-36}$ condition, and the effects on the Load was equivalent to the heat load condition at 31 °C. In contrast, the Load of feeling of physical fatigue was increased according to the degree of temperature step.

We first examined the effects of repeated temperature steps with environments of 26 °C, 31 °C, and 36 °C based on entering an environment of 26 °C. Primarily, a Bayesian state–space model was used to estimate the accumulated effects

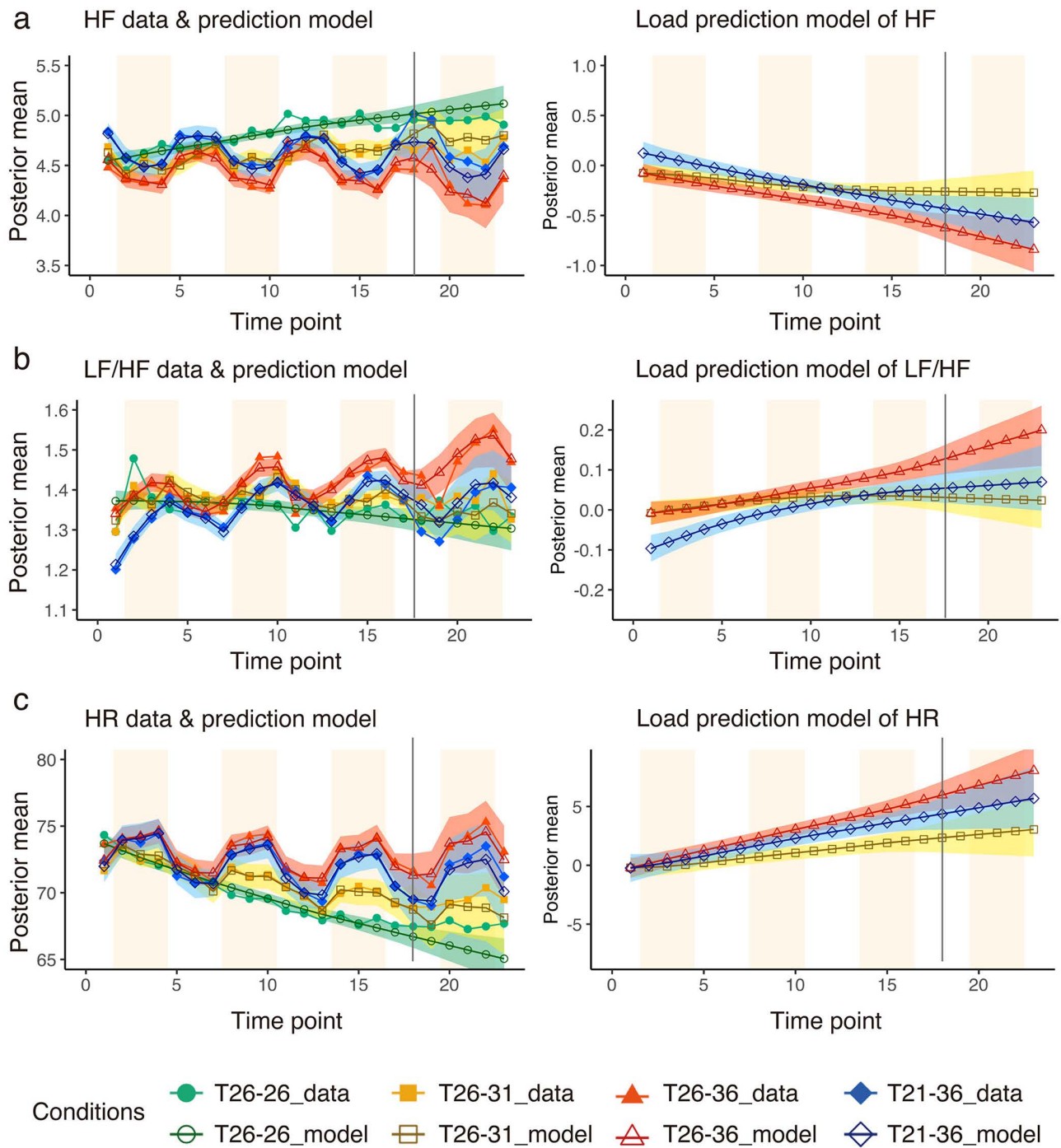

**Fig 5. Prediction of heart rate variability.** Results of (a) HF (high-frequency component of heart rate variability), (b) LF/HF (the ratio between low-frequency component and high-frequency component of heart rate variability), and (c) HR (heart rate) modelling. Model estimates to the left of the vertical black line are based on actual measurements at the corresponding time points, and models after the fourth set on the right side of the black vertical line are estimated using the measured values up to the third set. The colour band represents the 50% credible interval. The light orange background indicates Room B. The x-axis shows the time points, and the y-axis shows the measured value, calculated value, and predicted value of each index. Filled markers represent the measured values, and empty markers represent the estimated models.

(load) of repeated temperature steps by excluding the immediate effects of repeated temperature changes (environment) and changes that commonly occur under all conditions (Base). A heat load of 36 °C rather than 31 °C increases the Load on the body, including a decrease in HF, while an increase in HR, LF/HF, skin temperature, and feelings of fatigue (Figs 1 and 2).

To maintain vital activities, humans maintain body temperature homeostasis through various physiological mechanisms, such as dilation of skin blood vessels and heat dissipation through sweating [21]. When body temperature rises in a hot environment, the physiological load increases owing to thermoregulatory functions such as increased HR and dehydration caused by sweating [22]. This study showed that the Load term related to skin temperature and HR gradually increased with temperature steps with repeated exposure to the hot environments compared to that under control conditions, which are maintained at room temperature. This indicates that a 21 min stay in the room-temperature environment between heat exposure was insufficient to fully recover a load of thermal exposure owing to the hot environments and that the body accumulated a load owing to repeated thermal exposure [21,23].

Notably, the $T_{26-36}$ condition resulted in a higher increase in skin temperature than that under the other conditions, whereas the $T_{26-36}$, $T_{21-36}$, and $T_{26-31}$ conditions elicited higher HRs in this order. Since the skin temperature load increased under the $T_{26-36}$, $T_{21-36}$, and $T_{26-31}$ conditions, staying in an environment of 26 °C or 21 °C for the given minutes between heat exposure did not have a considerable effect in suppressing the increase in skin temperature load. Furthermore, the Loads of $T_{21-36}$ and $T_{26-31}$ have been suppressed more than those of $T_{26-36}$ because of attenuated responses in thermoregulation, i.e., cutaneous vasodilation and sweating [24]. The fact that the Load of $T_{26-36}$ was the highest suggests that the skin temperature load increase was more dependent on the absolute value of the environmental temperature than on the temperature steps, and the HR load was higher in the order of $T_{26-36}$, $T_{21-36}$, and $T_{26-31}$, suggesting that it was more dependent on the absolute value of skin temperature and environmental temperature than on the temperature steps.

Subsequently, the effect of the cooling environment was examined based on a heat load of 36 °C for repeated exposure, which is a high temperature. For skin temperature and LF/HF, all values under the $T_{21-36}$ condition started at a lower value than those under the other conditions, whereas HF started at a higher value because the $T_{21-36}$ condition started at 21 °C, which was lower than that for the other condition. Although $T_{21-36}$ had the largest range of temperature differences, the Loads for skin temperature, HF, and LF/HF fell within the $T_{26-31}$ range, even after repeated heat loads of 36 °C. It is likely that the body was cooled by the cooler environment at 21 °C, and the heat load on the body was reduced. Similar results were obtained in a previous study by moving participants from a high- to a low-room temperature environment in a single step test [15].

In this study, we examined the psychophysiological effects of repeated temperature exposure. In particular, when the temperature difference was 10 °C or more, the psychophysiological load was large, and fatigue could be visualised clearly using the Load term of the autonomic nerve activity index. Parasympathetic nerve activity decreases, and the relative sympathetic nerve activity increases in a state of fatigue [17]. This was reflected by the Load on the temperature steps in this study.

The LF/HF under the $T_{26-36}$ condition shows that the Load term (which was mildly increased until the second set) increased especially in the third and fourth sets, indicating that repeated exposure to heat accelerates the accumulation of physical fatigue. However, the Load term of subjective physical fatigue (as measured by the subjective psychological index) was higher in the first set and plateaued without fluctuation. Notably, subjective physical fatigue plateaued at an early stage of temperature step load, whereas loads on the autonomic nervous system increased with longer loading time. This suggests that monitoring autonomic nerve activity, which is an objective fatigue evaluation index, is important in preventing fatigue accumulation, rather than evaluations based on subjective fatigue.

The results of this study show that state–space models using Bayesian statistics estimate the current psychological and physical load that individuals receive from the environment and the accumulated future load. We showed that the prediction model yielded precise results for skin temperature, while there was a lack of overlap between the prediction

model and measured values for other parameters. There may be more parameters that should be included in the prediction model. Since sensitivity and vulnerability to thermal load may depend on gender, body size, age, illness, and other conditions, it is expected that models with such parameters added as hidden variables would be more accurate. Such a highly accurate model could visualise the amount of psychophysiological load on human subjects in real time and lead to the development of solutions to reduce and prevent risks to users from the environmental side. For example, if a wearable device can acquire biometric signals and precisely predict the Load on the body and mind in real time, it can alert the person to the risk. Furthermore, the device can encourage the person to move to a cooler place or to take an adequate rest. We could automatically control the air conditioning of the environment by using such signals from the device. Such solutions are expected to reduce excessive physical and mental strain.

## Limitation

In this study, the experiment was conducted in a precisely controlled laboratory. While the sample size (N = 28) was modest, our Bayesian state-space modeling approach was specifically chosen to leverage the rich information within each participant's time-series data. The robustness of this analysis was confirmed by successful model convergence and excellent goodness-of-fit (e.g., R-hat values approaching 1.0), ensuring the reliability of our findings. We acknowledge, however, that further work is needed to enhance the model's generalizability to real-world scenarios. Future studies should aim to test the model with larger sample sizes, incorporate additional parameters such as activity levels and sleep patterns, and utilize conditions that more closely reflect real-world environmental changes.

Elaborating on these future directions, the incorporation of additional parameters is particularly crucial for addressing a limitation of our predictive model. Specifically, while our forecasting model demonstrated good convergence and yielded precise results for skin temperature, there was a lack of overlap between the prediction model and measured values for other parameters. Since individual characteristics such as gender, body size, age, activity levels, sleep patterns, and health status can influence sensitivity and vulnerability to thermal load, it is expected that models with such parameters added as hidden variables would be more accurate.

Beyond these individual-specific factors, the model's applicability to real-world scenarios also requires considering environmental variables not examined in our controlled laboratory setting. For example, this study did not examine the effect of long-term temperature differential load on fatigue. Rather, we examined repeated exposure to temperature steps over a relatively short period of time, approximately every 21 min. In the real-world, people are exposed to steps in indoor and outdoor temperature as they enter and exit during daily activities and to temperature steps between day and night as well as daily temperature differences. Moreover, seasonal temperature variations have longer term temperature differential load. Further studies are needed on repeated temperature steps over longer periods. Although this study was a highly controlled laboratory experiment, the intervals of exposure to temperature steps and magnitude of temperature steps in real life are often unequal. It is also necessary to examine the effects of random exposure times in environments with random magnitudes of temperature steps. Furthermore, it is not known whether similar results can be obtained for temperature difference environments in winter since this experiment was conducted assuming a summer season. We focused on strict temperature control and not on humidity. Studies focusing on the effects of humidity steps on psychophysiological responses may provide further insights.

A major limitation of this study was that core body temperature could not be measured owing to the configuration of the experiment. It is necessary to measure the core body temperature in future experiments since it is a critical parameter offering useful information regarding the primary functioning of body processes.

## Conclusion

We modelled the psychophysiological effects of repeated temperature steps using a state–space model of Bayesian statistics. We differentiated and estimated the immediate effects of environmental changes from the accumulated effects of repeated

temperature steps. The results suggest that repeatedly entering a hot environment at 36 °C increases the Load on the skin temperature, autonomic nervous system, and psychological index. In this study, the parasympathetic index decreased when repeated exposure exceeded three or four steps. Interestingly, the results show that the Load on the body may be reduced by entering a low-temperature environment below room temperature after being in a high-temperature environment even when repeatedly exposed to a high-temperature environment. However, the Load of subjective physical fatigue increased according to the degree of temperature step. The method of estimation and prediction of accumulated effects of repeated temperature steps will be useful for future studies and for application to the real world with environmental stress.

## Supporting information

**S1 Table. Feeling of physical fatigue model parameters.**
(XLSX)

**S2 Table. Skin temperature model parameters.**
(XLSX)

**S3 Table. HF model parameters.**
(XLSX)

**S4 Table. LF/HF model parameters.**
(XLSX)

**S5 Table. HR model parameters.**
(XLSX)

**S6 Table. Feeling of physical fatigue prediction model parameters.**
(XLSX)

**S7 Table. Skin temperature prediction model parameters.**
(XLSX)

**S8 Table. HF prediction model parameters.**
(XLSX)

**S9 Table. LF/HF prediction model parameters.**
(XLSX)

**S10 Table. HR prediction model parameters.**
(XLSX)

**S11 Table. Raw data of each parameter.**
(XLSX)

## Acknowledgments

We are grateful to Marie Miyamoto and Yuka Takahashi for their assistance in conducting the research. We thank Editage (www.editage.jp) for English language editing.

## Author contributions

**Conceptualization:** Miho Iwasaki, Yusuke Morito, Kiyoshi Kuroi, Shota Hori, Yasuyoshi Watanabe.
**Data curation:** Miho Iwasaki, Yusuke Morito.

**Formal analysis:** Miho Iwasaki, Kyosuke Watanabe.

**Investigation:** Miho Iwasaki, Yusuke Morito, Yoko Sakata.

**Methodology:** Miho Iwasaki, Yusuke Morito, Kyosuke Watanabe, Shota Hori.

**Project administration:** Yasuyoshi Watanabe.

**Software:** Miho Iwasaki, Kyosuke Watanabe.

**Supervision:** Yasuyoshi Watanabe.

**Visualization:** Miho Iwasaki.

**Writing – original draft:** Miho Iwasaki, Yusuke Morito.

**Writing – review & editing:** Miho Iwasaki, Kyosuke Watanabe, Kei Mizuno, Kazunobu Okazaki, Yasuyoshi Watanabe.

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
