## [Decision Letter · Decision Letter 0]

2 Jul 2025

Dear Dr. Watanabe,

We look forward to receiving your revised manuscript.

Kind regards,

Neelu Jain Gupta, Ph.D.

Academic Editor

PLOS ONE

Journal Requirements:

“Y.W. received funding for this study from DAIKIN Industries, Ltd. Co.”

3. Please note that funding information should not appear in the Acknowledgments section or other areas of your manuscript. We will only publish funding information present in the Funding Statement section of the online submission form. Please remove any funding-related text from the manuscript. 

“I have read the journal's policy and the authors of this manuscript have the following competing interests: Y.W. received funding for this study from DAIKIN Industries, Ltd. Co. K.K., S.H., and Y.S. are employees of DAIKIN Industries, Ltd. Co. The other authors have declared that no competing interests exist.”

We note that you received funding from a commercial source: DAIKIN Industries, Ltd. Co.

“I have read the journal's policy and the authors of this manuscript have the following competing interests: Y.W. received funding for this study from DAIKIN Industries, Ltd. Co. K.K., S.H., and Y.S. are employees of DAIKIN Industries, Ltd. Co. The other authors have declared that no competing interests exist.”

We note that one or more of the authors are employed by a commercial company: DAIKIN Industries, Ltd. Co.

2) Please also provide an updated Competing Interests Statement declaring this commercial affiliation along with any other relevant declarations relating to employment, consultancy, patents, products in development, or marketed products, etc.  

Within your Competing Interests Statement, please confirm that this commercial affiliation does not alter your adherence to all PLOS ONE policies on sharing data and materials by including the following statement: ""This does not alter our adherence to  PLOS ONE policies on sharing data and materials.” (as detailed online in our guide for authors http://journals.plos.org/plosone/s/competing-interests) . If this adherence statement is not accurate and  there are restrictions on sharing of data and/or materials, please state these. Please note that we cannot proceed with consideration of your article until this information has been declared.

6. We note that you have indicated that there are restrictions to data sharing for this study. For studies involving human research participant data or other sensitive data, we encourage authors to share de-identified or anonymized data. However, when data cannot be publicly shared for ethical reasons, we allow authors to make their data sets available upon request. For information on unacceptable data access restrictions, please see http://journals.plos.org/plosone/s/data-availability#loc-unacceptable-data-access-restrictions. 

7. In this instance it seems there may be acceptable restrictions in place that prevent the public sharing of your minimal data. However, in line with our goal of ensuring long-term data availability to all interested researchers, PLOS’ Data Policy states that authors cannot be the sole named individuals responsible for ensuring data access (http://journals.plos.org/plosone/s/data-availability#loc-acceptable-data-sharing-methods).

**Additional Editor Comments:**

Linking daily temperature variations between indoors and outdoors to health is pertinent worldwide. The study requires significant adjustments because its current methodology is very constrained 1) sample size specially when it addresses psychophysiological burdens. The planned study is fine, however the physical exhaustion scale is too broad (0–100), making assessments more arbitrary. One reviewer has rightly suggested that tests such as as skin temperature, ECG, HR, and HRV, factors like activity and sleep (including latency, inertia, and duration of sleep) could be taken into account. I suggest you rework on methodological errors to improve current form of manuscript. Study limitations should be adequately mentioned under relevant headings.

Reviewers' comments:

Reviewer's Responses to Questions

**Comments to the Author**

1. Is the manuscript technically sound, and do the data support the conclusions?

Reviewer #1: Yes

Reviewer #2: Partly

2. Has the statistical analysis been performed appropriately and rigorously?

Reviewer #1: Yes

Reviewer #2: Yes

3. Have the authors made all data underlying the findings in their manuscript fully available?

Reviewer #1: Yes

Reviewer #2: Yes

4. Is the manuscript presented in an intelligible fashion and written in standard English?

Reviewer #1: Yes

Reviewer #2: Yes

Reviewer #1: Despite the insufficient sample size, the study design was sound and thoroughly examined. We recommend conducting this study over a longer period of time and with a larger sample size in order to examine the physiological and psychological impacts of several successive temperature increments.

Reviewer #2: The wordwide exposure of the humans exposed to daily temperature differences indoors and outdoors associated to health risks and fatigue is a thoughtful concept. The study needs phenomenal changes as the approach is quite limited/ narrow.

The study aimed to clarify the psychophysiological loads by repeated short-term temperature differences on n = 28 (14 males and 14 females) Japanese individuals. The study planned is good, but of very short duration as the data collected for each test is of only 183 minutes for 4 times. First the duration of the study should be more, may be at least for a week or so, secondly, the effects should be recorded not only in the test duration but also pre and post test, to proove the case. The scale of physical fatigue is quite large (0-100), making the psychological measurements more subjective.Besides the psychophysiological and physiological measurements like HR, HRV, skin temperature and ECG, parameters like activity and sleep (latency, inertia and sleep duration) should also be considered. The statistics has been thoroghly performed, but is too complex to understand. The comparisons between the temperature- controlled and the treated groups would have made the study/ statistics simple and easy to prove.

Overall the experimental design should be more elaborate in order to demonstrate good internal control and structure. However, the lack of contextual and participant-specific data, along with a few methodological omissions, has impacted the interpretability and external validity of the findings.

**Do you want your identity to be public for this peer review?** For information about this choice, including consent withdrawal, please see our Privacy Policy

Reviewer #1: No

Reviewer #2: No

---

## [Author Response · Author response to Decision Letter 1]

29 Aug 2025

Responses to Additional Editor Comments:

[Part 1]

Linking daily temperature variations between indoors and outdoors to health is pertinent worldwide. The study requires significant adjustments because its current methodology is very constrained 1) sample size specially when it addresses psychophysiological burdens.

[Response]

We sincerely thank the Editor for the thoughtful and constructive comments regarding our manuscript. As this study was conducted in a tightly controlled laboratory environment, our primary aim was to explore psychophysiological responses to short-term temperature changes under experimental conditions. We fully agree that further research with a larger sample size and in real-world environments will be essential to improve generalizability and assess broader health implications. To address this, we have added a Limitation section in the revised manuscript, where we discuss this issue in detail (see lines 418–430).

[Part 2]

The planned study is fine, however the physical exhaustion scale is too broad (0–100), making assessments more arbitrary.

[Response]

We thank the editor’s comment on the use of a 0–100 visual analogue scale (VAS) for assessing physical fatigue. The VAS is a widely used and validated tool for capturing subjective fatigue, as shown by Lee et al. (1991). Its continuous format allows for detecting subtle changes that may be missed by categorical scales, making it suitable for psychophysiological studies like ours. To minimize variability, we gave participants clear instructions with anchor definitions (0 = no fatigue, 100 = complete exhaustion) and ensured consistency across sessions. We have also revised the Methods section to better explain our use of the VAS (see lines 112–118).

[Part 3]

One reviewer has rightly suggested that tests such as as skin temperature, ECG, HR, and HRV, factors like activity and sleep (including latency, inertia, and duration of sleep) could be taken into account. I suggest you rework on methodological errors to improve current form of manuscript. Study limitations should be adequately mentioned under relevant headings.

[Response]

We thank the Editor for highlighting the importance of incorporating additional physiological and behavioral indicators such as skin temperature, ECG, HR, HRV, sleep parameters (latency, inertia, duration), and physical activity. We fully agree that these factors are essential for developing a more ecologically valid and comprehensive model of thermally induced fatigue in real-world settings. In the present study, our focus was to establish a baseline understanding of the psychophysiological burden under temperature step conditions in a controlled environment. Nevertheless, we acknowledge this limitation and have added corresponding statements in the new Limitation section of the revised manuscript to outline directions for future research (see lines 418–457).

Responses to Reviewer #1

Reviewer #1’s comments:

Despite the insufficient sample size, the study design was sound and thoroughly examined. We recommend conducting this study over a longer period of time and with a larger sample size in order to examine the physiological and psychological impacts of several successive temperature increments.

[Response]

We sincerely thank the reviewer for recognizing the strengths of our study design. We fully agree with the suggestion that conducting the study over a longer period and with a larger sample size would help clarify the physiological and psychological impacts of repeated temperature changes more robustly.

As this study was designed as a preliminary investigation conducted under controlled laboratory conditions, we focused on establishing a basic psychophysiological response model. In the revised manuscript, we have now included a Limitation section discussing this point and emphasizing the need for future studies with extended duration and broader sample populations (see lines 418–457).

Responses to Reviewer #2

Reviewer #2’s comments:

The worldwide exposure of the humans exposed to daily temperature differences indoors and outdoors associated to health risks and fatigue is a thoughtful concept. The study needs phenomenal changes as the approach is quite limited/narrow. The study aimed to clarify the psychophysiological loads by repeated short-term temperature differences on n = 28 (14 males and 14 females) Japanese individuals. The study planned is good, but of very short duration as the data collected for each test is of only 183 minutes for 4 times. First the duration of the study should be more, may be at least for a week or so, secondly, the effects should be recorded not only in the test duration but also pre and post test, to prove the case.

The scale of physical fatigue is quite large (0-100), making the psychological measurements more subjective.

Besides the psychophysiological and physiological measurements like HR, HRV, skin temperature and ECG, parameters like activity and sleep (latency, inertia and sleep duration) should also be considered.

The statistics has been thoroughly performed, but is too complex to understand. The comparisons between the temperature- controlled and the treated groups would have made the study/statistics simple and easy to prove.

Overall the experimental design should be more elaborate in order to demonstrate good internal control and structure. However, the lack of contextual and participant-specific data, along with a few methodological omissions, has impacted the interpretability and external validity of the findings.

[Response]

We thank the reviewer for the thorough review of our manuscript. We sincerely appreciate for recognizing our research concept as "thoughtful" and for providing numerous constructive comments.

We have addressed each of the concerns below and have made revisions throughout the manuscript based on your valuable feedback.

1. Study duration and measurement timing

We appreciate the reviewer’s thoughtful comments regarding the limited study duration and measurement design. This study was conducted under controlled laboratory conditions with the aim of exploring short-term psychophysiological responses to temperature steps. We fully agree that longer-term studies with pre- and post-assessments, as well as real-world observations, are essential for future work. We have added a new Limitation section to the manuscript addressing this issue (see lines 418–430).

2. Use of VAS for fatigue assessment

We thank the reviewer’s comment on the use of a 0–100 visual analogue scale (VAS) for assessing physical fatigue. The VAS is a widely used and validated tool for capturing subjective fatigue, as shown by Lee et al. (1991). Its continuous format allows for detecting subtle changes that may be missed by categorical scales, making it suitable for psychophysiological studies like ours. To minimize variability, we gave participants clear instructions with anchor definitions (0 = no fatigue, 100 = complete exhaustion) and ensured consistency across sessions. We have also revised the Methods section to better explain our use of the VAS (see lines 112–118).

3. Consideration of additional parameters such as sleep and activity

We thank the reviewer for highlighting the importance of including additional physiological and behavioral factors such as activity and sleep (latency, inertia, and duration). We agree that incorporating these variables would enhance the ecological validity of future models. In this study, our focus was to establish a psychophysiological load model within a controlled setting. We have now noted the absence of these variables and the need for more comprehensive assessments in future studies in the Limitation section (see lines 431–437).

4. Statistical methodology

We appreciate the reviewer’s feedback regarding the complexity of the statistical approach. To address our primary goal—capturing the temporal dynamics of psychophysiological load—we employed a Bayesian state-space model. This method is particularly suited for time-series data with repeated measures, where each observation is autocorrelated with prior states.

Unlike conventional methods that assume independence between time points, the state-space model distinguishes between the latent internal state (e.g., underlying fatigue) and random observation noise, allowing us to track how participants’ internal states evolve over time. This approach provides a more accurate assessment of intervention effects than simple group-level comparisons at discrete time points.

We also included a temperature-stable condition (26°C–26°C) as a control group for comparison.

---

## [Decision Letter · Decision Letter 1]

13 Oct 2025

Estimating Psychophysiological Loads by Repeated Temperature Steps on Humans using a State–Space Model

PONE-D-25-06325R1

Dear Dr. Watanabe,

We’re pleased to inform you that your manuscript has been judged scientifically suitable for publication and will be formally accepted for publication once it meets all outstanding technical requirements.

Kind regards,

Neelu Jain Gupta, Ph.D.

Academic Editor

PLOS ONE

Additional Editor Comments (optional):

Reviewers' comments:

Reviewer's Responses to Questions

**Comments to the Author**

Reviewer #1: All comments have been addressed

Reviewer #2: All comments have been addressed

2. Is the manuscript technically sound, and do the data support the conclusions?

Reviewer #1: Yes

Reviewer #2: Yes

3. Has the statistical analysis been performed appropriately and rigorously?

Reviewer #1: Yes

Reviewer #2: Yes

4. Have the authors made all data underlying the findings in their manuscript fully available?

Reviewer #1: Yes

Reviewer #2: Yes

5. Is the manuscript presented in an intelligible fashion and written in standard English?

Reviewer #1: Yes

Reviewer #2: Yes

Reviewer #1: The study effectively addresses a timely issue of repeated temperature changes and their psychophysiological impact using a robust Bayesian state–space modeling approach.

The experimental design is well controlled and methodologically sound, though the small sample size (n=28) and short duration (183 minutes) limit broader generalization.

i recommend authors the Inclusion of parameters such as sleep, activity, and circadian factors can enhance ecological validity in future studies.

The Bayesian modeling is appropriate and insightful, but interpretations should remain cautious given laboratory constraints and limited real-world validation.

Overall, the manuscript is clearly written, logically structured, and transparent about its limitations, with constructive directions for future research.

Reviewer #2: These findings provide valuable insights that could contribute to the development of effective fatigue management strategies and the establishment of best practices to reduce adverse health effects from sudden and uncontrollable environmental temperature changes in daily life. I commend the authors for this important preliminary work and would encourage them to further strengthen it by addressing the limitations noted in the current study, while also underscoring the need for future investigations with longer durations and more diverse populations.

**Do you want your identity to be public for this peer review?** For information about this choice, including consent withdrawal, please see our Privacy Policy

Reviewer #1: No

Reviewer #2: **Yes: ** Shalie Malik

---

## [Editor Report · Acceptance letter]

PONE-D-25-06325R1

PLOS ONE

Dear Dr. Watanabe,

I'm pleased to inform you that your manuscript has been deemed suitable for publication in PLOS ONE. Congratulations! Your manuscript is now being handed over to our production team.

Kind regards,

on behalf of

Dr. Neelu Jain Gupta

Academic Editor

PLOS ONE